# Prevention of Occupational Strain: Can Psychological Empowerment and Organizational Commitment Decrease Dissatisfaction and Intention to Quit?

**DOI:** 10.3390/jcm7110450

**Published:** 2018-11-20

**Authors:** Marta Llorente-Alonso, Gabriela Topa

**Affiliations:** Department of Social and Organizational Psychology, UNED (National University of Distance Education), 28040 Madrid, Spain; gtopa@psi.uned.es

**Keywords:** occupational strain, psychological empowerment, organizational commitment, satisfaction, intention to quit, multiple mediation

## Abstract

In the present study, the demands-control-support model has served as the basis for the assessment of occupational strain. This model has been used as a predictor of health problems. It has also been associated with organizational outcomes and behaviors. The purpose of this study is to relate job demands and resources with job satisfaction and intention to quit the union. We intend to test a multiple mediation model with psychological empowerment and union commitment as mediator variables. The investigation was carried out with 953 delegates of a Spanish trade union (healthcare professionals). We collected 401 questionnaires. Multiple mediation analyses were performed with bootstrapping techniques using the SPSS PROCESS macro. The results underlined the effects of multiple mediation of empowerment and commitment in the relation between resources and job satisfaction. This mediation was also observed in the relation between resources and intention to quit. The lack of relation between demands and satisfaction or intention to quit is of interest. In the presence of adequate resources, delegates are empowered and committed to their union, which leads to lower dissatisfaction and lower rates of quitting. This study advises organizations to give greater importance to motivational and attitudinal factors to attenuate occupational strain.

## 1. Introduction

Healthcare professionals are an important focus of interest for research centered on the assessment of occupational strain (a combination between high psychologic demands of an occupation and low decision latitude) [1] and the characteristics of the working environment, such as job demands and work resources [2,3,4]. Nevertheless, only a small amount of studies focus on the working characteristics of trade union delegates. The conflicts inherent to their union work, the variety of the tasks they perform, and the growing problems in the work centers generate high job demands in delegates. As a result, they suffer a work overload in their role [5]. In addition, healthcare professionals’ participation in the trade union is another reason for work stress. Most of them have little knowledge or training in collective negotiations or labor topics. They do not belong to professions related to labor law. Therefore, the role of the union delegate within company committees requires knowledge to mediate in work relations. Competence and control are also necessary [6].

### 1.1. The Demands–Resources Models as A Framework for the Assessment of Occupational Strain

The demands–control model of Karasek [1] and its subsequent expansion to the job demands-control-support model of Johnson and Hall [7] have frequently been used in research for the assessment of work characteristics and stressors [8]. The demands-control model proposes that the interactions between the levels of job demands and control generate different types of psychosocial experiences at work: High-stress jobs, low-stress jobs, and active and passive jobs. The extension of this model with the dimension of social support proposes a buffering effect of the negative effects of high stress at work [7]. Jobs in which high stress is experienced are characterized by high job demands and low control. Low-stress jobs are located in the opposite extreme. Passive jobs are characterized by low demand and low control. Active jobs include jobs with high demand and high control. The extension of this model with the dimension of social support proposes a buffering effect of the negative effects of high stress at work [7].

The abovementioned models have been used as predictors of health problems, such as cardiovascular risk [7], burnout syndrome [4], diseases of the digestive system [9] (p. 135), and metabolic syndrome [10,11]. Another study has shown that the job demands-control-support model was a significant predictor of depressive symptoms (except job demands) [12]. They have also been associated with organizational outcomes and behaviors. For example, high-stress jobs have been linked to higher levels of mobbing [13], less psychological empowerment of the workers, and lower job satisfaction [2].

Schaubroeck and Fink also found that the interaction between demands, support, and control predicted symptoms of health, sick leave, organizational commitment, and satisfaction with the supervisor [14]. Van Yperen and Hagedoorn suggested that control at work reduced fatigue in high-demand jobs [3]. Job stress is not necessarily present in high-demand job settings. Job stress can be avoided when control at work compensates for the high demands [15]. Regarding perceived organizational support, Chou, Hecker, and Martin found that it is positively related to job satisfaction [16]. In addition, the staff in health organizations with consistent levels of resources and demands have lower intentions to quit [17].

In the reviewed literature, there is a relationship between job demands and resources and organizational outcomes. However, most studies have focused on healthcare professionals. This research aims to study the occupational strain of healthcare professionals’ union delegates and to determine their influence on job satisfaction and intention to quit the trade union:
H1A: Job demands will be directly and negatively related to the union delegates’ job satisfaction.H1B: Work resources will be directly and positively related to union delegates’ job satisfaction.H2A: Job demands will be directly and positively related to intention to quit the union.H2B: Work resources will be directly and negatively related to intention to quit the union.

### 1.2. Psychological Empowerment

Psychological empowerment is a motivational concept of self-efficacy to enable workers to take initiative [18]. It refers to a series of cognitive processes that modify people’s subjective self-perceptions and their environment or context. Empowerment consists of intrinsic motivation for the task manifested by means of four conditions: Meaning, competence, impact, and choice or self-determination [19]. Spreitzer has highlighted the importance of perceptions in the interpretation of the work environment in which individuals feel empowered. In this study, the author positively relates sociopolitical support, the area of control, access to information, and participatory climate to psychological empowerment [20]. The dimensions of meaning and competence are related to stress at work and job satisfaction [21]. Along these same lines, Laschinger, Finegan, Shamian, and Almost suggested that nurses employed in work environments with a high level of control feel more psychologically empowered [2].

Psychological empowerment plays a relevant role as a motivational factor. This function allows it to modify the relation between the characteristics of the work environment and the organizational outcomes. Psychological empowerment along with structural empowerment culminate in positive organizational retention outcomes, such as job satisfaction [22]. A meta-analysis showed that psychological empowerment and job satisfaction are significantly positively correlated [23]. Other authors associated empowerment with the intention to quit [24].

### 1.3. Union Commitment

From an attitudinal perspective, organizational commitment is defined as “a state in which an individual identifies with a certain organization and its goals, and wants to maintain membership in order to facilitate these goals” [25]. Focusing on this theoretical approach, Gordon, Philpot, Burt, Thompson, and Spiller developed the measure of union commitment [26]. It is made up of four dimensions: Union loyalty, willingness to work for its goals, responsibility towards the organization, and belief in unionism.

Davis found that union commitment directly increases union members’ job satisfaction [27]. Cohen conducted an investigation whose participants were members of a nursing union in Israel. In this study, union loyalty was strongly associated with job satisfaction and intention to quit the union [28]. Other studies have found that employees have more positive attitudes towards work and greater organizational commitment when demands and resources are high [29]. Various investigations have revealed the relation between psychological empowerment and organizational commitment [19,30]. More psychologically empowered employees are more likely to feel higher levels of commitment towards their organization [31]. Higher levels of psychological empowerment have been associated with higher organizational commitment and greater job satisfaction. Nevertheless, differences were found in the levels of these variables as a function of the country in which the data were collected [32].

The abovementioned investigations have presented separately the importance of the role of empowerment and commitment to achieving appropriate organizational results. In addition, commitment has been studied as an attitudinal consequence of empowerment [33]. This research proposes that in the presence of an adequate social environment, psychological empowerment and union commitment exert a joint action to achieve greater satisfaction and fewer intentions to quit (see Figure 1):
H1C: The relation between demands and satisfaction will be multiply mediated by psychological empowerment and union commitment. H1D: The relation between resources and satisfaction will be multiply mediated by psychological empowerment and union commitment.H2C: The relation between demands and intention to quit will be multiply mediated by psychological empowerment and union commitment. H2D: The relation between resources and intention to quit will be multiply mediated by psychological empowerment and union commitment. 

## 2. Method

### 2.1. Participants

The study was carried out with a sample of union delegates (*N* = 401) of a Spanish trade union. This trade union is composed of 953 union delegates, who were our study population. All of them were health professionals. At the time of collection of the data, they were mostly registered nurses, although other categories can also belong to the union. 523 delegates (51.88%) worked full-time for the union. By contrast, 430 (45.12%) worked mostly for their contracting company, making use of union hours to serve as representatives on company committees and personnel boards.

The sample consisted of 99.5% of registered nurses (399 nurses and 2 physiotherapists). Mean age was 47.82 years (SD = 7.86). Of the total of the sample, 70.3% were women, 86.5% had a fixed-term contract, and 76.3% of the participants worked full-time for the union. The mean number of years as a union delegate in the organization was 8.18 years (SD = 7.18).

### 2.2. Procedure

We requested the permission of the presidency of the union to access the participants. The questionnaire was sent to 953 union delegates in February, 2017. We informed them of the objective of the research and of the confidential nature of the data. The participants gave their consent. It was clearly stated that the questionnaire should be answered according to their employment relationship with the union. The questions did not refer to the center or hospital in which they work. The deadline established to fill in the questionnaires was one month after starting to collect the data. The final sample was made up of 401 people.

### 2.3. Instruments

#### 2.3.1. Psychosocial Characteristics of the Job: Demands and Resources (Support and Control)

To appraise the psychosocial characteristics of the job, we used a reduced minimal version of the job content questionnaire (JCQ) [9], validated in Spanish [34]. This version is made up of three subscales. The internal consistency indexes obtained by these authors in the three subscales were: 0.74 for Job Demands, 0.87 for Support, and 0.74 for Control at Work.

Job Demands is made up of six items, and the reliability of the scale in this study was α = 0.77. Support consists of nine items, α = 0.93. Control is made up of seven items, α = 0.82. Example items are: “My job requires working very hard” (demands), “My supervisor pays attention to what I say” (support), and “My job requires me to be creative” (control). The Likert response scale of the items ranged from 1 (totally disagree) to 5 (totally agree).

#### 2.3.2. Psychological Empowerment

We used the Spanish adaptation [35] of the psychological empowerment in the workplace scale of Spreitzer [36]. The instrument consists of 12 items distributed in the subscales of Meaning, Competence, Self-determination, and Impact. Example items are: “The work I do in the union is important to me,” and “I trust my ability to do the work.” The Likert response scale of the items ranged from 1 (totally disagree) to 5 (totally agree). The internal consistency of this questionnaire has been reported in other investigations [32,36]. In this study, the reliability of the scale was α = 0.90. Another study confirmed the convergent validity of the questionnaire [36]. Each dimension contributes to a general construct of psychological empowerment.

#### 2.3.3. Union Commitment

To appraise union commitment, we used two subscales (Union Loyalty and Willingness to Engage in Union Work) of the questionnaire developed and validated by Kelloway, Catano, and Southwell [37]. We decided not to use the subscale Responsibility toward the Union because its items evaluate a procedure for the processing of complaints that is not included in the Spanish labor legislation. The Union Loyalty scale has been used in other studies and has a high level of internal consistency [38]. We also used the subscale Willingness to Engage in Union Work due to the adequate adaptation of the items to the characteristics and functioning of the union. In this research, reliability was α = 0.87 for the Loyalty subscale and 0.76 for subscale Willingness to Engage in Union Work. Examples of items are: “I am proud to be part of this union” and “If they ask me, I would run for elected office in the union”. The Likert response scale of the items ranged from 1 (totally disagree) to 5 (totally agree).

#### 2.3.4. Job Satisfaction

We used the Job Satisfaction subscale of the Michigan organizational assessment questionnaire [39]. It consists of three items, one of them with a reversed score. An example item is: “All in all, I am satisfied with my work in the union.” It was rated on a 5-point Likert response scale, ranging from 1 (strongly disagree) to 5 (strongly agree). The reliability of the scale in this study was α = 0.68. Other studies have found values ranging from 0.67 to 0.95 [40].

#### 2.3.5. Intention to Quit the Union

We used a three-item scale developed by Aryee and Wah [41]. The response options ranged from 1 (very unlikely) to 5 (very likely) for the questions “Is there any likelihood that you will quit the union?” and “How likely is it for you to quit being a union delegate during your current contract with your company/administration?”, whereas for the question “Do you have any intention to quit the union?”, the range was 1 (strongly disagree) to 5 (strongly agree). This measure in this study showed a reliability of α = 0.83.

## 3. Results

### 3.1. Descriptive Analysis and Correlations

The means, standard deviations, and correlations of all the variables of the study are presented in Table 1.

### 3.2. Hypothesis Testing

To test the hypotheses of the study, various multiple mediation analyses were conducted. We used bootstrapping techniques with the macro PROCESS for SPSS (model Nr. 6) designed by Andrew Hayes [42].

Regarding job satisfaction, corresponding to Hypothesis 1, we conducted three multiple mediation analyses. In each analysis, one of the work characteristics (demands, support, or control) acted as the independent variable and the other two as covariates. To test multiple mediation, we introduced psychological empowerment and union commitment as mediators.
To test 1A and 1C, firstly, we carried out the analysis with job demands. The direct effect of job demands on job satisfaction was nonsignificant (c‘ = −0.005, *p* = 0.91). The indirect effects of the mediation of psychological empowerment and union commitment between job demands and job satisfaction were nonsignificant, as they included the value 0 in the confidence intervals with a 95% level. Hypotheses 1A and 1C were not supported.Secondly, continuing with the study of job satisfaction and to test Hypotheses 1B and 1D, we analyzed support. The direct effect of support on job satisfaction was nonsignificant (c‘ = 0.064, *p* = 0.069). Regarding the indirect effects, only those corresponding to the isolated mediation of empowerment and to the multiple mediation of empowerment and union commitment were significant. As there were two significant effects, we compared the indirect effects to determine which of them had more statistical significance for the model. The comparison of the isolated effect of the mediation of psychological empowerment with the multiple effect was nonsignificant (95% IC [−0.02,0.05]). Therefore, it cannot be concluded that the isolated mediation of empowerment is higher than the multiple mediation.Thirdly, we related the variable control to job satisfaction, finding a nonsignificant direct effect (c‘ = 0.0013, *p* = 0.98). The indirect effect of the isolated mediation of union commitment was also nonsignificant. Nevertheless, the indirect effects of the simple mediation of empowerment and the multiple mediation of empowerment and union commitment were significant. Again, two indirect, significant effects coexist, so a comparative analysis was performed. The superiority of the isolated effect of empowerment over the multiple effect could not be confirmed (95% IC [−0.10,0.16]). Therefore, Hypothesis 1B was not supported, whereas Hypothesis 1D was confirmed (See Table 2 and Figure 2).

The next group of hypotheses focused on intention to quit. They were contrasted following the same methodology as with job satisfaction.
Firstly, to test Hypotheses 2A and 2C, we related job demands to intention to quit, finding a nonsignificant direct effect (c’ = 0.087, *p* = 0.309). No indirect effect was significant. These results led to rejecting Hypotheses 2a and 2c.Secondly, to test Hypotheses 2B and 2D, we studied support at work. Its direct effect on intention to quit was nonsignificant (c’ = 0.024, *p* = 0.718). Regarding the indirect effects, the effect corresponding to the multiple mediation of empowerment and union commitment was significant. The isolated effects of the mediation of empowerment and of union commitment were nonsignificant.Thirdly, we tested control. Its direct effect on intention to quit was nonsignificant (c’ = 0.034, *p* = 0.758). The only significant indirect effect was that corresponding to multiple mediation. These latter results of the variables support and control led to the rejection of Hypothesis 2B, but also to the confirmation of Hypothesis 2D (See Table 3 and Figure 2).

## 4. Discussion

In the present study, the demands-control-support model served as the basis for the assessment of the occupational strain. We aimed to relate job demands and resources (support and control) to union delegates’ job satisfaction and intention to quit the union. We tested a multiple mediation model with psychological empowerment and union commitment as mediator variables. The results partially confirmed the proposed hypotheses.

Firstly, we highlight the effects of multiple mediation exerted by psychological empowerment and union commitment in the relation between work resources and job satisfaction. This multiple mediation is also observed in the relation between resources and intention to quit the union. These mediation effects are linked to the absence of a direct relation between work resources and job satisfaction or intention to quit. These results are in the line of those of Bakker, Demerouti, and De Boer [43]. In their study, they stressed the role of work resources as a unique predictor of organizational commitment. They emphasized the action of commitment as a mediator between work resources and the frequency of sick leave from work. In addition, the findings that Seibert, Wang, and Coutright developed in their meta-analysis are expanded [33]. They explained that psychological empowerment is related to a broad range of organizational outcomes, such as job satisfaction and intention to quit. It is also related to organizational commitment as an attitudinal factor. They stressed sociopolitical support as an antecedent of psychological empowerment. Our results agree with the suggested relations. We highlight the role of union commitment as a mediator of the relation between work resources and organizational outcomes, along with psychological empowerment.

On another hand, Laschinger, Finegan, Shamian, and Almost found that when increasing control of work, nurses expressed higher levels of job satisfaction, more organizational commitment, and lower intention to quit, and they felt more psychologically empowered [2]. This study highlights the existence of full mediation as a novelty regarding the investigations reviewed. The effect of work resources on the outcomes is entirely due to the existence of the mediators’ empowerment and commitment. We underline the importance of the motivational and attitudinal factors in the relation between job characteristics and organizational outcomes.

In addition, we observed that in the relation between work resources and job satisfaction, there were two simultaneous, significant mediation effects: Simple mediation of psychological empowerment and multiple mediation of empowerment and commitment. When comparing the two indirect effects, it was found that the simple effect of empowerment was not greater than the multiple effect. These results reflect the importance of empowerment in the relation between work resources and job satisfaction. However, it should be noted that in the presence of union commitment, the relation improves.

Secondly, there is a lack of relation between job demands and job satisfaction, and intention to quit. There is no direct or indirect relation through the multiple mediation of empowerment and commitment. Other investigations have agreed about the importance of the levels of resources, regardless of the job demands, to obtain appropriate organizational outcomes [2]. Karasek explained that occupational stress can be improved by increasing control at work, regardless of the job demands [1]. This shows the importance of work resources to achieve organizational outcomes, independently of the demands.

### 4.1. Limitations

Given the cross-sectional nature of this study, no causal relations can be established. As the main threat to external validity, we consider a potential Hawthorne effect [44]. When they feel they are being evaluated, participants tend to react to the environment and respond according to their perception of social desirability. This effect may be aggravated if someone in a high position of the organization requested their collaboration in the study. On the other hand, the questionnaire is long and may have tired the participants. In addition, the participation rate was low.

Regarding the model used, reference to the conceptualization of the demands is relevant. Van der Doef and Maes [8] explained that in the model of Karasek and Theorell, the central component of demands was the workload. Nevertheless, in jobs like nursing, work stressors and demands may be more related to the interaction with patients than to the workload. In the case of this study, the lack of relevance of job demands could be related to the fact that in the model used, interaction with affiliates or coworkers was not considered. Therefore, it is important to highlight that the model may not be the best for studying occupational stress in trade unionists. On the other hand, the choice of an ineffective model weakens the relationship, so that relationships could be even stronger than we observed.

In addition, the lack of studies on union delegates may affect this investigation. Most of the exposed theory focuses on healthcare professionals, but not on union delegates.

## 5. Conclusions

The results of the present study suggest that organizations should grant greater importance to motivational and attitudinal factors, oriented to implementing internal practices that involve empowering the employees and increasing their commitment. The absence of psychological empowerment may imply the perception of lack of competence. According to our theoretical model, it would generate a lower commitment to the organization, higher rates of quitting, and a lower satisfaction.

This study opens the door to new investigations that grant more importance to the role of motivational and attitudinal factors in the relation between work environment and organizational outcomes. In addition, this study emphasizes a process of multiple mediation of psychological empowerment and union commitment. We recommend performing longitudinal studies to examine the successive process of resources, empowerment, commitment, and organizational outcomes. In addition, we recommend undertaking additional studies to further understanding of the union delegates’ occupational strain.

## Figures and Tables

**Figure 1 jcm-07-00450-f001:**
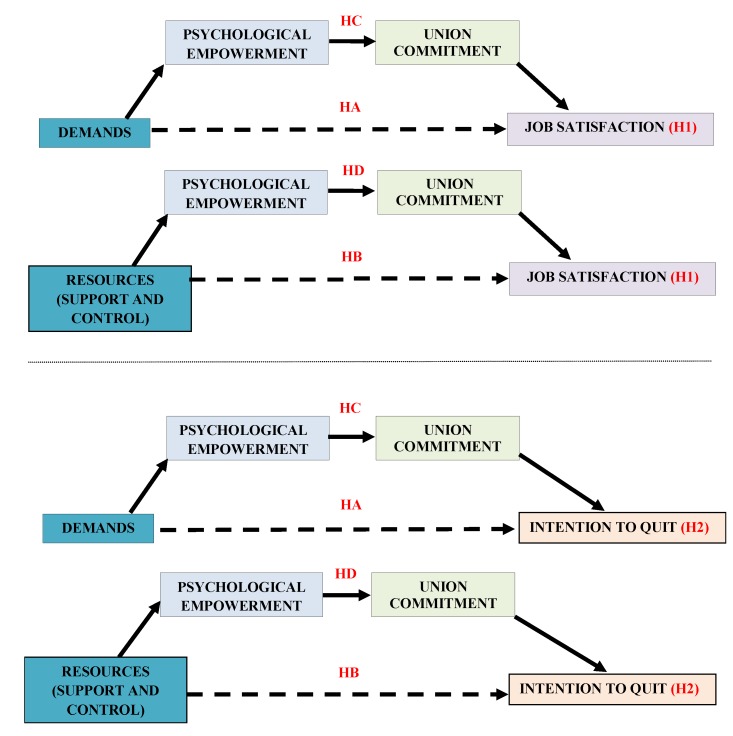
Hypotheses and proposed model. Note: HA: Job demands will be directly and negatively related to job satisfaction/intention to quit. HB: Work resources will be directly and positively related to job satisfaction/intention to quit. HC: The relation between demands and satisfaction/intention to quit will be multiply mediated by psychological empowerment and union commitment. HD: The relation between resources and satisfaction/intention to quit will be multiply mediated by psychological empowerment and union commitment.

**Figure 2 jcm-07-00450-f002:**
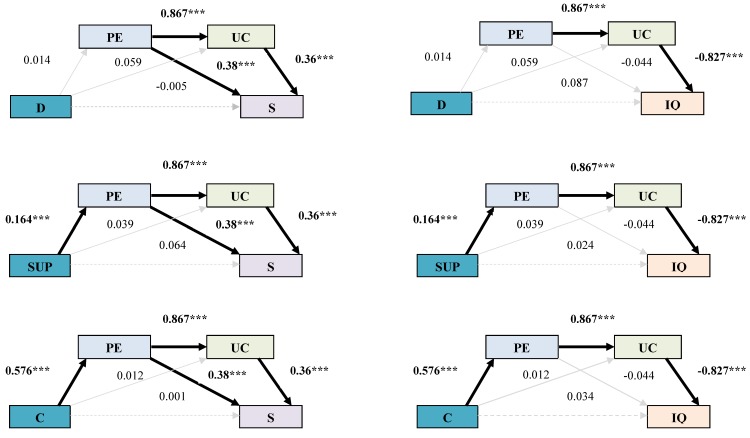
Final model. Nonstandardized B Coefficients and statistical significance. *** *p* < 0.001. Note: D = Demands; PE = psychological Empowerment; UC = Union Commitment; S = Satisfaction; Sup = Support; C = Control; IQ = Intention to quit.

**Table 1 jcm-07-00450-t001:** Correlations, means, and standard deviations of study variables.

Variables	Mean	SD	1	2	3	4	5	6
1. Resources (Support)	4.25	0.75	-					
2. Resources (Control)	4.01	0.58	0.45 **	-				
3. Demands	4.16	0.56	0.21 **	0.45 **	-			
4. Psychological empowerment	4.11	0.57	0.48 **	0.69 **	0.33 **	-		
5. Union commitment	3.99	0.68	0.41 **	0.55 **	0.30 **	0.77 **	-	
6. Job satisfaction	4.40	0.64	0.39 **	0.48 **	0.24 **	0.67 **	0.67 **	-
7. Intention to quit the union	2.29	1.02	−0.20 **	−0.27 **	−0.11 **	−0.41 **	−0.54 **	−0.50 **

Notes: * *p* < 0.05, ** *p* < 0.01. *N* = 401.

**Table 2 jcm-07-00450-t002:** Results of mediation test of psychological empowerment and union commitment between job demands-resources and job satisfaction.

	Coefficients B	SE	*t*	Coefficients B	Boot SE	95%CI
**(H1a) Demands–Satisfaction**	−0.005	0.045	−0.110			
Indirect effect 1: Demands–PE–Satisfaction				0.005	0.018	[−0.030,0.044]
**Indirect effect 2: (H1c) Demands–PE–UC–Satisfaction**				0.004	0.016	[−0.023,0.040]
Indirect effect 3: Demands–UC–Satisfaction				0.021	0.017	[−0.011,0.059]
**(H1b) Support–Satisfaction**	0.064	0.035	1.81			
Indirect effect 1: Support–PE–Satisfaction				0.063 **	0.017	[0.033,0.104]
**Indirect effect 2 (H1d): Support– PE–UC–Satisfaction**				0.052 **	0.014	[0.030,0.086]
Indirect effect 3: Support–UC–Satisfaction				0.014	0.015	[−0.014,0.046]
**(H1b) Control–Satisfaction**	0.001	0.058	0.022			
Indirect effect 1: Control–PE–Satisfaction				0.220 **	0.046	[0.134,0.319]
**Indirect effect 2 (H1d): Control–PE–UC–Satisfaction**				0.183 **	0.040	[0.115,0.281]
Indirect effect 3: Control–UC–Satisfaction				0.004	0.025	[−0.046,0.055]

Note: *N* = 401; SE = Standard error; CI = Confidence Intervals; Boot SE = Standard error of bootstrap. PE = Psychological Empowerment; UC = Union Commitment; Sample size bootstrap for indirect effects = 10,000; **, *p* < 0.01.

**Table 3 jcm-07-00450-t003:** Results of mediation test of psychological empowerment and union commitment between demands-resources and intention to quit the union.

	Coefficients B	SE	*t*	Coefficients B	Boot SE	95%CI
**(H1a) Demands–IQ**	0.087	0.085	1.01			
Indirect effect 1: Demands–PE–IQ				−0.0006	0.007	[−0.024,0.009]
**Indirect effect 2: (H1c) Demands–PE–UC–IQ**				−0.010	0.036	[−0.090,0.055]
Indirect effect 3: Demands–UC–IQ				−0.049	0.038	[−0.124,0.023]
**(H1b) Support–IQ**	0.024	0.066	0.360			
Indirect effect 1: Support–PE–IQ				−0.007	0.023	[−0.058,0.037]
**Indirect effect 2 (H1d): Support–PE–UC–IQ**				−0.118 **	0.029	[−0.187,−0.068]
Indirect effect 3: Support–UC–IQ				−0.032	0.034	[−0.103,0.032]
**(H1b) Control–IQ**	0.034	0.110	0.307			
Indirect effect 1: Control–PE–IQ				−0.025	0.080	[−0.183,0.130]
**Indirect effect 2 (H1d): Control–PE–UC–IQ**				−0.413 **	0.074	[−0.578,−0.285]
Indirect effect 3: Control–UC–IQ				−0.010	0.058	[−0.125,0.102]

Note: *N* = 401; SE = Standard error; CI = Confidence Intervals; Boot SE = Standard error of bootstrap. PE = Psychological Empowerment; UC = Union Commitment; IQ = Intention to quit. Sample size bootstrap for indirect effects = 10,000; **, *p* < 0.01.

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
