# Peer review of "Prevention of Occupational Strain: Can Psychological Empowerment and Organizational Commitment Decrease Dissatisfaction and Intention to Quit?"

_jcm, 2018, doi:10.3390/jcm7110450_

Reviewer 1 Report

Karasek's demand-control model was born in the '70s to interpret the workers' discomfort of the assembly line. it is very sensitive to workloads and to the hierarchical position. it is certainly not the best model for studying stress in trade unionists whose work is based rather on a concept of exchange between the effort provided and the results achieved. Siegrist's effort / reward imbalance model would have been much better suited to describe this population.

The authors should correctly recognize this limit of their study design. On the other hand, the choice of an ineffective model weakens the relationship, so they could say that the observed relationships are, in reality, even stronger than what is observed.

 The authors should indicate a very important data to understand the study: what was the time that the union delegates dedicated to their trade union activity, and what was the time dedicated to nursing activity? Those who enjoy "union posting", that is, who work full-time in the union, are very different from those who work full-time in the hospital and also participate in trade union life.

The large number of people interviewed would suggest that they were mostly nurses with a union card. In this case there is perhaps a methodological problem: a questionnaire (demand / control / support) that investigates hospital work has been put in relation with one that investigates the commitment with the union.

 I personally believe that the intention to leave the union derives from how the union is led, not from how the delegate lives his/her work.

 This methodological uncertainty is reflected in the text. In the abstract there is talk of "intention to quit" without specifying that the study is aimed at the intention to quit the union. Readers think that workers want to leave the nursing profession. In the discussion, uncertainty is even more serious. The authors compare their study on the intention to leave the union, with studies in which there was intention to leave work. The statement at L. 297 “the lack of a relation between job demands and job satisfaction and intention to quit is interesting” is wrong and must be corrected into “the lack of a relation between job demands and job satisfaction and intention to quit union is …” I think the latter relationship is less interesting than the former, and authors must correctly explain why.

 Lines 58-65. The references used to indicate the validity of the demand-control model are rather dated. The model has been used very recently, for example to demonstrate the association between stress and metabolic syndrome in a longitudinal study [Garbarino S, Magnavita N. Work stress and metabolic syndrome in police officers. A prospective study. PLoSOne, 2015 Dec 7; 10 (12):e0144318. doi: 10.1371/journal.pone.0144318], or to show association of stress with mental illnesses [Garbarino S, Cuomo G, Chiorri C, Magnavita N. Association of work-related stress with mental health problems in a special police force. BMJOpen 2013 Jul 19;3(7). doi:pii: e002791. 10.1136/bmjopen-2013-002791]. Authors should mention these and other similar studies conducted on health care workers.

 The participation rate was low. This is a limitation of the study.

 Author Response

Thank you very much for your work and for giving us the opportunity of resubmitting the paper.

We have taken into account your suggestions and we have made the following changes.

We have highlighted the modified text in yellow.

 Point 1: Karasek's demand-control model was born in the '70s to interpret the workers' discomfort of the assembly line. it is very sensitive to workloads and to the hierarchical position. it is certainly not the best model for studying stress in trade unionists whose work is based rather on a concept of exchange between the effort provided and the results achieved. Siegrist's effort / reward imbalance model would have been much better suited to describe this population.

The authors should correctly recognize this limit of their study design. On the other hand, the choice of an ineffective model weakens the relationship, so they could say that the observed relationships are, in reality, even stronger than what is observed.

 Response 1:

·         The model is not the best for studying occupational stress in trade unionists (line 370-375).

 Point 2: The authors should indicate a very important data to understand the study: what was the time that the union delegates dedicated to their trade union activity, and what was the time dedicated to nursing activity? Those who enjoy "union posting", that is, who work full-time in the union, are very different from those who work full-time in the hospital and also participate in trade union life.

 Response 2:

·         (Line 153) The time that the union delegates dedicated to their trade union activity: (76.3% of the participants worked full time for the union).

 Point 3: The large number of people interviewed would suggest that they were mostly nurses with a union card. In this case there is perhaps a methodological problem: a questionnaire (demand / control / support) that investigates hospital work has been put in relation with one that investigates the commitment with the union.

 I personally believe that the intention to leave the union derives from how the union is led, not from how the delegate lives his/her work.

 This methodological uncertainty is reflected in the text. In the abstract there is talk of "intention to quit" without specifying that the study is aimed at the intention to quit the union. Readers think that workers want to leave the nursing profession. In the discussion, uncertainty is even more serious. The authors compare their study on the intention to leave the union, with studies in which there was intention to leave work.

 Response 3:

·         The study refers to work done as a union delegate (line 161). The participants knew this information when they answered the questionnaire.

·         We compare our study on the intention to leave the union, with studies in which there was intention to leave work. We add a limitation (line 377) and a future line of research (392)

·         Intention to quit the union (abstract line 14)

 Point 4: The statement at L. 297 “the lack of a relation between job demands and job satisfaction and intention to quit is interesting” is wrong and must be corrected into “the lack of a relation between job demands and job satisfaction and intention to quit union is …” I think the latter relationship is less interesting than the former, and authors must correctly explain why.

 Response 4:

·         We delete that the relationship between demands and satisfaction is interesting (line 350). We add a limitation: “The lack of relevance of job demands could be related to the model used…” (lines 370-375).

 Point 5: Lines 58-65. The references used to indicate the validity of the demand-control model are rather dated. The model has been used very recently, for example to demonstrate the association between stress and metabolic syndrome in a longitudinal study [Garbarino S, Magnavita N. Work stress and metabolic syndrome in police officers. A prospective study. PLoSOne, 2015 Dec 7; 10 (12):e0144318. doi: 10.1371/journal.pone.0144318], or to show association of stress with mental illnesses [Garbarino S, Cuomo G, Chiorri C, Magnavita N. Association of work-related stress with mental health problems in a special police force. BMJOpen 2013 Jul 19;3(7). doi:pii: e002791. 10.1136/bmjopen-2013-002791]. Authors should mention these and other similar studies conducted on health care workers.

 Response 5:

·         We add new references (line 57).

Point 6: The participation rate was low. This is a limitation of the study.

Response 6:

·         The participation rate was low (Line 365).

Reviewer 2 Report

This cross-sectional study addresses an important and relevant issue regarding the occupational strain of healthcare professionals' union delegates, and is aimed to determine their influence on job satisfaction and intention to quit the trade union. This has the potential to be an interesting line of research. However, the authors did not assess the risk of bias; particularly, I think that the paper suffers an high risk of selection bias; such bias might affect the research outcome and might  distort the results. There is no information (e.g. demographic data) about the population (n. 953) from which the participants (n. 401)  come.  The authors should re-analyse the data considering the high risk of bias. The above mentioned lack of methodology does not allow to attribute significance to the results of the study. Findings and conclusion are invaluable because potentially affected by bias and, consequently, distorted.

Author Response

Thank you very much for your recommendation.

We have taken into account your comments and we have provided information about the population. Please, see lines 145-154. Moreover. As suggested by Podsakoff and colleagues (2003), procedural remedies were used to control and counteract common method variance. To reduce evaluation apprehension and prevent response distortion, the respondents’ anonymity was protected with respect to their employer, respondents were advised that there were no right or wrong answers, and they were asked to answer questions as honestly as possible. Moreover, we seek to control response consistencies by counterbalancing the question order in the survey by placing the measures of the dependent variables before those of the independent variables (Harrison et al., 1996). Furthermore, the study variables were measured with validated scales, which can mitigate measurement error and thereby, decrease common method bias (Spector, 1987). 

Besides, we have checked the manuscript for spelling and grammatical errors. We have highlighted the modified text in yellow.

Reviewer 3 Report

Thank you for the opportunity to review your manuscript. I found the paper interesting to read and your analysis was appropriate and relevant to your study objectives. My suggestions are that you review the feedback (comments and in the yellow highlighted text) that I have provided in the attached manuscript. I believe they will strengthen your paper. As importantly, there were numerous grammatical and typographical errors that will detract the reader from understanding your study's important findings.

Author Response

 Thank you very much for your work and for giving us the opportunity of resubmitting the paper.

We have taken into account your suggestions and we have made the following changes. We have highlighted the modified text in yellow.

·         We have added a definition of occupational strain.

·         We have added information about the population and the sample.

·         We have added a citation about the Hawthorne effect.

·         We have modified grammatical errors and we have carefully revised the manuscript again.

Round  2

Reviewer 1 Report

The manuscript has been improved

Reviewer 2 Report

The manuscript has been improved; the authors have sufficiently clarified the critical aspects of the previous version.

Reviewer 3 Report

I am happy support the acceptance of this article to the journal editors